# Optimistic Concurrency Control for Distributed Unsupervised Learning

**Xinghao Pan**[1] **Joseph Gonzalez**[1] **Stefanie Jegelka**[1] **Tamara Broderick**[1,2] **Michael I. Jordan**[1,2]
[1]Department of Electrical Engineering and Computer Science, and [2]Department of Statistics
University of California, Berkeley
Berkeley, CA USA 94720
{xinghao,jegonzal,stefje,tab,jordan}@eecs.berkeley.edu

## Abstract

Research on distributed machine learning algorithms has focused primarily on one of two extremes—algorithms that obey strict concurrency constraints or algorithms that obey few or no such constraints. We consider an intermediate alternative in which algorithms optimistically assume that conflicts are unlikely and if conflicts do arise a conflict-resolution protocol is invoked. We view this "optimistic concurrency control" paradigm as particularly appropriate for large-scale machine learning algorithms, particularly in the unsupervised setting. We demonstrate our approach in three problem areas: clustering, feature learning and online facility location. We evaluate our methods via large-scale experiments in a cluster computing environment.

## 1 Introduction

The desire to apply machine learning to increasingly larger datasets has pushed the machine learning community to address the challenges of distributed algorithm design: partitioning and coordinating computation across the processing resources. In many cases, when computing statistics of iid data or transforming features, the computation factors according to the data and coordination is only required during aggregation. For these *embarrassingly parallel* tasks, the machine learning community has embraced the map-reduce paradigm, which provides a template for constructing distributed algorithms that are fault tolerant, scalable, and *easy to study*.

However, in pursuit of richer models, we often introduce statistical dependencies that require more sophisticated algorithms (e.g., collapsed Gibbs sampling or coordinate ascent) which were developed and studied in the *serial* setting. Because these algorithms iteratively transform a global state, parallelization can be challenging and often requires frequent and complex coordination.

Recent efforts to distribute these algorithms can be divided into two primary approaches. The **mutual exclusion** approach, adopted by [1] and [2], guarantees a *serializable* execution preserving the theoretical properties of the serial algorithm but at the expense of parallelism and costly locking overhead. Alternatively, in the **coordination-free** approach, proposed by [3] and [4], processors communicate frequently without coordination minimizing the cost of contention but leading to stochasticity, data-corruption, and requiring potentially complex analysis to prove algorithm correctness.

In this paper we explore a third approach, **optimistic concurrency control** (OCC) [5] which offers the performance gains of the coordination-free approach while at the same time ensuring a serializable execution and preserving the theoretical properties of the serial algorithm. Like the coordination-free approach, OCC exploits the infrequency of data-corrupting operations. However, instead of allowing occasional data-corruption, OCC detects data-corrupting operations and applies correcting computation. As a consequence, OCC automatically ensures correctness, and the analysis is only necessary to guarantee optimal scaling performance.

We apply OCC to distributed nonparametric unsupervised learning—including but not limited to clustering—and implement distributed versions of the DP-Means [6], BP-Means [7], and online facility location (OFL) algorithms. We demonstrate how to analyze OCC in the context of the DP-Means algorithm and evaluate the empirical scalability of the OCC approach on all three of the proposed algorithms. The primary contributions of this paper are:

1. Concurrency control approach to distributing unsupervised learning algorithms.

2. Reinterpretation of online nonparametric clustering in the form of facility location with approximation guarantees.

3. Analysis of optimistic concurrency control for unsupervised learning.

4. Application to feature modeling and clustering.

## 2   Optimistic Concurrency Control

Many machine learning algorithms iteratively transform some global state (e.g., model parameters or variable assignment) giving the illusion of serial dependencies between each operation. However, due to sparsity, exchangeability, and other symmetries, it is often the case that many, *but not all*, of the state-transforming operations can be computed concurrently while still preserving **serializability**: the equivalence to some serial execution where individual operations have been reordered.

This opportunity for serializable concurrency forms the foundation of distributed database systems. For example, two customers may concurrently make purchases exhausting the inventory of unrelated products, but if they try to purchase the same product then we may need to serialize their purchases to ensure sufficient inventory. One solution (*mutual exclusion*) associates locks with each product type and forces each purchase of the same product to be processed serially. This might work for an unpopular, rare product but if we are interested in selling a popular product for which we have a large inventory the serialization overhead could lead to unnecessarily slow response times. To address this problem, the database community has adopted **optimistic concurrency control** (OCC) [5] in which the system tries to satisfy the customers requests without locking and corrects transactions that could lead to negative inventory (e.g., by forcing the customer to checkout again).

Optimistic concurrency control exploits situations where most operations can execute concurrently without conflicting or violating serialization invariants. For example, given sufficient inventory the order in which customers are satisfied is immaterial and concurrent operations can be executed serially to yield the same final result. However, in the rare event that inventory is nearly depleted two concurrent purchases may not be serializable since the inventory can never be negative. By shifting the cost of concurrency control to rare events we can admit more costly concurrency control mechanisms (e.g., re-computation) in exchange for an efficient, simple, coordination-free execution for the majority of the events.

Formally, to apply OCC we must define a set of **transactions** (i.e., operations or collections of operations), a mechanism to detect when a transaction violates serialization invariants (i.e., cannot be executed concurrently), and a method to correct (e.g., rollback) transactions that violate the serialization invariants. Optimistic concurrency control is most effective when the cost of validating concurrent transactions is small and conflicts occur infrequently.

Machine learning algorithms are ideal for optimistic concurrency control. The conditional independence structure and sparsity in our models and data often leads to sparse parameter updates substantially reducing the chance of conflicts. Similarly, symmetry in our models often provides the flexibility to reorder serial operations while preserving algorithm invariants. Because the models encode the dependency structure, we can easily detect when an operation violates serial invariants and correct by rejecting the change and rerunning the computation. Alternatively, we can exploit the semantics of the operations to resolve the conflict by accepting a modified update. As a consequence OCC allows us to easily construct provably correct and efficient distributed algorithms without the need to develop new theoretical tools to analyze complex non-deterministic distributed behavior.

## 2.1 The OCC Pattern for Machine Learning

Optimistic concurrency control can be distilled to a simple pattern (meta-algorithm) for the design and implementation of distributed machine learning systems. We begin by evenly partitioning $N$ data points (and the corresponding computation) across the $P$ available processors. Each processor maintains a replicated view of the global state and *serially* applies the learning algorithm as a sequence of operations on its assigned data and *the global state*. If an operation mutates the global state in a way that preserves the serialization invariants then the operation is accepted locally and its effect on the global state, if any, is eventually replicated to other processors.

However, if an operation could potentially conflict with operations on other processors then it is sent to a unique serializing processor where it is rejected or corrected and the resulting global state change is eventually replicated to the rest of the processors. Meanwhile the originating processor either tentatively accepts the state change (if a rollback operator is defined) or proceeds as though the operation has been deferred to some point in the future.

While it is possible to execute this pattern asynchronously with minimal coordination, for simplicity we adopt the bulk-synchronous model of [8] and divide the computation into *epochs*. Within an epoch $t$, $b$ data points $\mathcal{B}(p, t)$ are evenly assigned to each of the $P$ processors. Any state changes or serialization operations are transmitted at the end of the epoch and processed before the next epoch. While potentially slower than an asynchronous execution, the bulk-synchronous execution is deterministic and can be easily expressed using existing systems like Hadoop or Spark [9].

# 3 OCC for Unsupervised Learning

Much of the existing literature on distributed machine learning algorithms has focused on classification and regression problems, where the underlying model is continuous. In this paper we apply the OCC pattern to machine learning problems that have a more discrete, combinatorial flavor—in particular unsupervised clustering and latent feature learning problems. These problems exhibit symmetry via their invariance to both data permutation and cluster or feature permutation. Together with the sparsity of interacting operations in their existing serial algorithms, these problems offer a unique opportunity to develop OCC algorithms.

The K-means algorithm provides a paradigm example; here the inferential goal is to partition the data. Rather than focusing solely on K-means, however, we have been inspired by recent work in which a general family of K-means-like algorithms have been obtained by taking Bayesian nonparametric (BNP) models based on combinatorial stochastic processes such as the Dirichlet process, the beta process, and hierarchical versions of these processes, and subjecting them to *small-variance asymptotics* where the posterior probability under the BNP model is transformed into a cost function that can be optimized [7]. The algorithms considered to date in this literature have been developed and analyzed in the serial setting; our goal is to explore distributed algorithms for optimizing these cost functions that preserve the structure and analysis of their serial counterparts.

## 3.1 OCC DP-Means

We first consider the *DP-means* algorithm (Alg. 1) introduced by [6]. Like the K-means algorithm, DP-Means alternates between updating the cluster assignment $z_i$ for each point $x_i$ and recomputing the centroids $\mathcal{C} = \{\mu_k\}_{k=1}^K$ associated with each clusters. However, DP-Means differs in that the number of clusters is not fixed a priori. Instead, if the distance from a given data point to all existing cluster centroids is greater than a parameter $\lambda$, then a new cluster is created. While the second phase is trivially parallel, the process of introducing clusters in the first phase is inherently serial. However, clusters tend to be introduced infrequently, and thus DP-Means provides an opportunity for OCC.

In Alg. 3 we present an OCC parallelization of the DP-Means algorithm in which each iteration of the serial DP-Means algorithm is divided into $N/(Pb)$ bulk-synchronous epochs. The data is evenly partitioned $\{x_i\}_{i \in \mathcal{B}(p,t)}$ across processor-epochs into blocks of size $b = |\mathcal{B}(p, t)|$. During each epoch $t$, each processor $p$ evaluates the cluster membership of its assigned data $\{x_i\}_{i \in \mathcal{B}(p,t)}$ using the cluster centers $\mathcal{C}$ from the previous epoch and optimistically proposes a new set of cluster centers $\hat{\mathcal{C}}$. At the end of each epoch the proposed cluster centers, $\hat{\mathcal{C}}$, are *serially* validated using Alg. 2.

**Algorithm 1:** Serial DP-means

**Input**: data $\{x_i\}_{i=1}^N$, threshold $\lambda$
$\mathcal{C} \leftarrow \emptyset$
**while** *not converged* **do**
    **for** *i = 1 to N* **do**
        $\mu^* \leftarrow \operatorname{argmin}_{\mu \in \mathcal{C}} \|x_i - \mu\|$
        **if** $\|x_i - \mu^*\| > \lambda$ **then**
            $z_i \leftarrow x_i$
            $\mathcal{C} \leftarrow \mathcal{C} \cup x_i$        // New cluster
        **else** $z_i \leftarrow \mu^*$        // Use nearest
    **for** $\mu \in \mathcal{C}$ **do** // Recompute Centers
        $\mu \leftarrow \texttt{Mean}(\{x_i \,|\, z_i = \mu\})$

**Output**: Accepted cluster centers $\mathcal{C}$

---

**Algorithm 2:** `DPValidate`

**Input**: Set of proposed cluster centers $\hat{\mathcal{C}}$
$\mathcal{C} \leftarrow \emptyset$
**for** $x \in \hat{\mathcal{C}}$ **do**
    $\mu^* \leftarrow \operatorname{argmin}_{\mu \in \mathcal{C}} \|x - \mu\|$
    **if** $\|x_i - \mu^*\| < \lambda$ **then** // Reject
        $\texttt{Ref}(x) \leftarrow \mu^*$    // Rollback Assgs
    **else** $\mathcal{C} \leftarrow \mathcal{C} \cup x$        // Accept

**Output**: Accepted cluster centers $\mathcal{C}$

---

**Algorithm 3:** Parallel DP-means

**Input**: data $\{x_i\}_{i=1}^N$, threshold $\lambda$
**Input**: Epoch size $b$ and $P$ processors
**Input**: Partitioning $\mathcal{B}(p,t)$ of data $\{x_i\}_{i \in \mathcal{B}(p,t)}$ to
        processor-epochs where $b = |\mathcal{B}(p,t)|$
$\mathcal{C} \leftarrow \emptyset$
**while** *not converged* **do**
    **for** *epoch t = 1 to N/(Pb)* **do**
        $\hat{\mathcal{C}} \leftarrow \emptyset$  // New candidate centers
        **for** $p \in \{1, \ldots, P\}$ **do in parallel**
            // Process local data
            **for** $i \in \mathcal{B}(p,t)$ **do**
                $\mu^* \leftarrow \operatorname{argmin}_{\mu \in \mathcal{C}} \|x_i - \mu\|$
                // Optimistic Transaction
                **if** $\|x_i - \mu^*\| > \lambda$ **then**
                    $z_i \leftarrow \texttt{Ref}(x_i)$
                    $\hat{\mathcal{C}} \leftarrow \hat{\mathcal{C}} \cup x_i$
                **else** $z_i \leftarrow \mu^*$  // Always Safe
        // Serially validate clusters
        $\mathcal{C} \leftarrow \mathcal{C} \cup \texttt{DPValidate}(\hat{\mathcal{C}})$
    **for** $\mu \in \mathcal{C}$ **do** // Recompute Centers
        $\mu \leftarrow \texttt{Mean}(\{x_i \,|\, z_i = \mu\})$

**Output**: Accepted cluster centers $\mathcal{C}$

Figure 1: The Serial DP-Means algorithm and distributed implementation using the OCC pattern.

The validation process accepts cluster centers that are not covered by (i.e., not within $\lambda$ of) already accepted cluster centers. When a cluster center is rejected we update its reference to point to the already accepted center, thereby correcting the original point assignment.

## 3.2 OCC Facility Location

The DP-Means objective turns out to be equivalent to the classic Facility Location (FL) objective: $J(\mathcal{C}) = \sum_{x \in X} \min_{\mu \in \mathcal{C}} \|x - \mu\|^2 + \lambda^2 |\mathcal{C}|$, which selects the set of cluster centers (facilities) $\mu \in \mathcal{C}$ that minimizes the shortest distance $\|x - \mu\|^2$ to each point (customer) $x$ as well as the penalized cost of the clusters $\lambda^2 |\mathcal{C}|$. However, while DP-Means allows the clusters to be arbitrary points (e.g., $\mathcal{C} \in \mathbb{R}^D$), FL constrains the clusters to be points $\mathcal{C} \subseteq \mathcal{F}$ in a set of candidate locations $\mathcal{F}$. Hence, we obtain a link between combinatorial Bayesian models and FL allowing us to apply algorithms with known approximation bounds to Bayesian inspired nonparametric models. As we will see in Section 4, our OCC algorithm provides constant-factor approximations for both FL and DP-means.

Facility location has been studied intensely. We build on the *online* facility location (OFL) algorithm described by Meyerson [10]. The OFL algorithm processes each data point $x$ serially in a single pass by either adding $x$ to the set of clusters with probability $\min(1, \min_{\mu \in \mathcal{C}} \|x - \mu\|^2 / \lambda^2)$ or assigning $x$ to the nearest existing cluster. Using OCC we are able to construct a distributed OFL algorithm (Alg. 4) which is nearly identical to the OCC DP-Means algorithm (Alg. 3) but which provides strong approximation bounds. The OCC OFL algorithm differs only in that clusters are introduced and validated stochastically—the validation process ensures that the new clusters are accepted with probability equal to the serial algorithm.

## 3.3 OCC BP-Means

*BP-means* is an algorithm for learning collections of latent binary *features*, providing a way to define groupings of data points that need not be mutually exclusive or exhaustive like clusters.

| **Algorithm 4:** Parallel OFL | **Algorithm 5:** OFLValidate |
|---|---|
| **Input**: Same as DP-Means<br>**for** *epoch t = 1 to N/(Pb)* **do** $\hat{\mathcal{C}} \leftarrow \emptyset$<br>    **for** $p \in \{1, \ldots, P\}$ **do in parallel**<br>        **for** $i \in \mathcal{B}(p, t)$ **do**<br>            $d \leftarrow \min_{\mu \in \mathcal{C}} \|x_i - \mu\|$<br>            **with probability** $\min \{d^2, \lambda^2\} / \lambda^2$<br>                $\hat{\mathcal{C}} \leftarrow \hat{\mathcal{C}} \cup (x_i, d)$<br><br>    $\mathcal{C} \leftarrow \mathcal{C} \cup \texttt{OFLValidate}(\hat{\mathcal{C}})$<br>**Output**: Accepted cluster centers $\mathcal{C}$ | **Input**: Set of proposed cluster centers $\hat{\mathcal{C}}$<br>$\mathcal{C} \leftarrow \emptyset$<br>**for** $(x, d) \in \hat{\mathcal{C}}$ **do**<br>    $d^* \leftarrow \min_{\mu \in \mathcal{C}} \|x - \mu\|$<br>    **with probability** $\min \{d^{*2}, d^2\} / d^2$<br>        $\mathcal{C} \leftarrow \mathcal{C} \cup x$       // Accept<br>**Output**: Accepted cluster centers $\mathcal{C}$ |

Figure 2: The OCC algorithm for Online Facility Location (OFL).

As with serial DP-means, there are two phases in serial BP-means (Alg. 6). In the first phase, each data point $x_i$ is labeled with binary assignments from a collection of features ($z_{ik} = 0$ if $x_i$ doesn't belong to feature $k$; otherwise $z_{ik} = 1$) to construct a representation $x_i \approx \sum_k z_{ik} f_k$. In the second phase, parameter values (the feature means $f_k \in \hat{\mathcal{C}}$) are updated based on the assignments. The first step also includes the possibility of introducing an additional feature. While the second phase is trivially parallel, the inherently serial nature of the first phase combined with the infrequent introduction of new features points to the usefulness of OCC in this domain.

The OCC parallelization for BP-means follows the same basic structure as OCC DP-means. Each transaction operates on a data point $x_i$ in two phases. In the first, analysis phase, the optimal representation $\sum_k z_{ik} f_k$ is found. If $x_i$ is not well represented (i.e., $\|x_i - \sum_k z_{ik} f_k\| > \lambda$), the difference is proposed as a new feature in the second validation phase. At the end of epoch $t$, the proposed features $\{f_i^{new}\}$ are serially validated to obtain a set of accepted features $\tilde{\mathcal{C}}$. For each proposed feature $f_i^{new}$, the validation process first finds the optimal representation $f_i^{new} \approx \sum_{f_k \in \tilde{\mathcal{C}}} z_{ik} f_k$ using *newly accepted features*. If $f_i^{new}$ is not well represented, the difference $f_i^{new} - \sum_{f_k \in \tilde{\mathcal{C}}} z_{ik} f_k$ is added to $\tilde{\mathcal{C}}$ and accepted as a new feature.

Finally, to update the feature means, let $F$ be the $K$-row matrix of feature means. The feature means update $F \leftarrow (Z^T Z)^{-1} Z^T X$ can be evaluated as a single transaction by computing the sums $Z^T Z = \sum_i z_i z_i^T$ (where $z_i$ is a $K \times 1$ column vector so $z_i z_i^T$ is a $K \times K$ matrix) and $Z^T X = \sum_i z_i x_i^T$ in parallel.

We present the pseudocode for the OCC parallelization of BP-means in Appendix A.

## 4  Analysis of Correctness and Scalability

In contrast to the coordination-free pattern in which scalability is trivial and correctness often requires strong assumptions or holds only in expectation, the OCC pattern leads to simple proofs of correctness and challenging scalability analysis. However, in many cases it is preferable to have algorithms that are correct and probably fast rather than fast and possibly correct. We first establish serializability:

**Theorem 4.1** (Serializability). *The distributed DP-means, OFL, and BP-means algorithms are serially equivalent to DP-means, OFL and BP-means, respectively.*

The proof (Appendix B) of Theorem 4.1 is relatively straightforward and is obtained by constructing a permutation function that describes an equivalent serial execution for each distributed execution. The proof can easily be extended to many other machine learning algorithms.

Serializability allows us to easily extend important theoretical properties of the serial algorithm to the distributed setting. For example, by invoking serializability, we can establish the following result for the OCC version of the online facility location (OFL) algorithm:

**Theorem 4.2.** *If the data is randomly ordered, then the OCC OFL algorithm provides a constant-factor approximation for the DP-means objective. If the data is adversarially ordered, then OCC OFL provides a log-factor approximation to the DP-means objective.*

The proof (Appendix B) of Theorem 4.2 is first derived in the serial setting then extended to the distributed setting through serializability. In contrast to divide-and-conquer schemes, whose approximation bounds commonly depend *multiplicatively* on the number of levels [11], Theorem 4.2 is unaffected by distributed processing and has no communication or coarsening tradeoffs. Furthermore, to retain the same factors as a batch algorithm on the full data, divide-and-conquer schemes need a large number of preliminary centers at lower levels [11, 12]. In that case, the communication cost can be high, since all proposed clusters are sent at the same time, as opposed to the OCC approach. We address the communication overhead (the number of rejections) for our scheme next.

**Scalability** The scalability of the OCC algorithms depends on the number of transactions that are rejected during validation (i.e., the rejection rate). While a general scalability analysis can be challenging, it is often possible to gain some insight into the asymptotic dependencies by making simplifying assumptions. In contrast to the coordination-free approach, we can still *safely* apply OCC algorithms in the absence of a scalability analysis or when simplifying assumptions do not hold.

To illustrate the techniques employed in OCC scalability analysis we study the DP-Means algorithm, whose scalability limiting factor is determined by the number of points that must be serially validated. We show that the communication cost only depends on the number of clusters and processing resources and does not directly depend on the number of data points. The proof is in Appendix C.

**Theorem 4.3** (DP-Means Scalability). *Assume $N$ data points are generated iid to form a random number ($K_N$) of well-spaced clusters of diameter $\lambda$: $\lambda$ is an upper bound on the distances within clusters and a lower bound on the distance between clusters. Then the expected number of* serially *validated points is bounded above by $Pb + \mathbf{E}[K_N]$ for $P$ processors and $b$ points per epoch.*

Under the separation assumptions of the theorem, the number of clusters present in $N$ data points, $K_N$, is exactly equal to the number of clusters found by DP-Means in $N$ data points; call this latter quantity $k_N$. The experimental results in Figure 3 suggest that the bound of $Pb + k_N$ may hold more generally beyond the assumptions above. Since the master must process at least $k_N$ points, the overhead caused by rejections is $Pb$ and independent of $N$.

# 5 Evaluation

For our experiments, we generated synthetic data for clustering (DP-means and OFL) and feature modeling (BP-means). The cluster and feature proportions were generated nonparametrically as described below. All data points were generated in $\mathbb{R}^{16}$ space. We fixed threshold parameter $\lambda = 1$.

**Clustering:** The cluster proportions and indicators were generated simultaneously using the stick-breaking procedure for Dirichlet processes—'sticks' are 'broken' on-the-fly to generate new clusters as necessary. For our experiments, we used a fixed concentration parameter $\theta = 1$. Cluster means were sampled $\mu_k \sim N(0, I_{16})$, and data points were generated at $x_i \sim N(\mu_{z_i}, \frac{1}{4}I_{16})$.

**Feature modeling:** We use the stick-breaking procedure of [13] to generate feature weights. Unlike with Dirichlet processes, we are unable to perform stick-breaking on-the-fly with Beta processes. Instead, we generate enough features so that with high probability ($> 0.9999$) the remaining non-generated features will have negligible weights ($< 0.0001$). The concentration parameter was also fixed at $\theta = 1$. We generated feature means $f_k \sim N(0, I_{16})$ and data points $x_i \sim N(\sum_k z_{ik} f_k, \frac{1}{4}I_{16})$.

## 5.1 Simulated experiments

To test the efficiency of our algorithms, we simulated the first iteration (one complete pass over all the data, where most clusters / features are created and thus greatest coordination is needed) of each algorithm in MATLAB. The number of data points, $N$, was varied from 256 to 2560 in intervals of 256. We also varied $Pb$, the number of data points processed in one epoch, from 16 to 256 in powers of 2. For each value of $N$ and $Pb$, we empirically measured $k_N$, the number of accepted clusters /

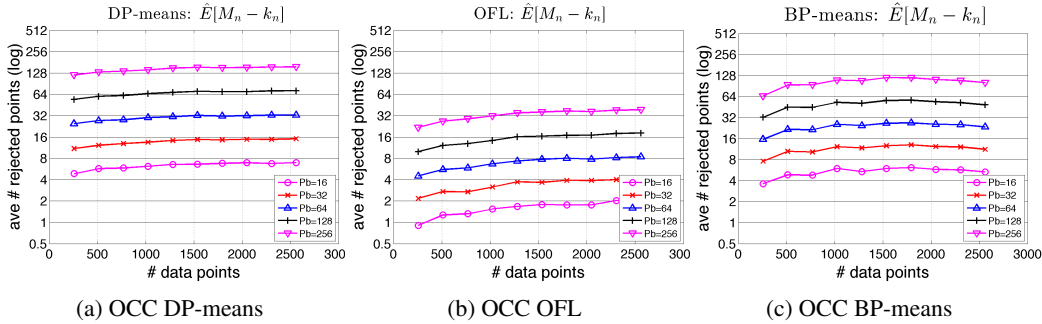

Figure 3: Simulated distributed DP-means, OFL and BP-means: expected number of data points proposed but not accepted as new clusters / features is independent of size of data set.

features, and $M_N$, the number of proposed clusters / features. This was repeated 400 times to obtain the empirical average $\hat{\mathbb{E}}[M_N - k_N]$ of the number of rejections.

For OCC DP-means, we observe $\hat{\mathbb{E}}[M_N - k_N]$ is bounded above by $Pb$ (Fig. 3a), and that this bound is independent of the data set size, even when the assumptions of Thm 4.3 are violated. (We also verified that similar empirical results are obtained when the assumptions are not violated; see Appendix C.) The same behavior is observed for the other two OCC algorithms (Fig. 3b and Fig. 3c).

## 5.2 Distributed implementation and experiments

We also implemented[1] the distributed algorithms in Spark [9], an open-source cluster computing system. The DP-means and BP-means algorithms were initialized by pre-processing a small number of data points (1/16 of the first $Pb$ points)—this reduces the number of data points sent to the master on the first epoch, while still preserving serializability of the algorithms. Our Spark implementations were tested on Amazon EC2 by processing a fixed data set on 1, 2, 4, 8 m2.4xlarge instances. Ideally, to process the same amount of data, an algorithm and implementation with perfect scaling would take half the runtime on 8 machines as it would on 4, and so on. The plots in Figure 4 shows this comparison by dividing all runtimes by the runtime on one machine.

**DP-means**: We ran the distributed DP-means algorithm on $2^{27} \approx 134M$ data points, using $\lambda = 2$. The block size $b$ was chosen to keep $Pb = 2^{23} \approx 8M$ constant. The algorithm was run for 5 iterations (complete pass over all data in 16 epochs). We were able to get perfect scaling (Figure 4a) in all but the first iteration, when the master has to perform the most synchronization of proposed centers.

**OFL**: The distributed OFL algorithm was run on $2^{20} \approx 1M$ data points, using $\lambda = 2$. Unlike DP-means and BP-means, OFL is a single-pass algorithm and we did not perform any initialization clustering. The block size $b$ was chosen such that $Pb = 2^{16} \approx 66K$ data points are processed each epoch, which gives us 16 epochs. Figure 4b shows that we get no scaling in the first epoch, where all $Pb$ data points are sent to the master. Scaling improves in the later epochs, as the master's workload decreases with fewer proposals but the workers' workload increases with more centers.

**BP-means**: Distributed BP-means was run on $2^{23} \approx 8M$ data points, with $\lambda = 1$; block size was chosen such that $Pb = 2^{19} \approx 0.5M$ is constant. Five iterations were run, with 16 epochs per iteration. As with DP-means, we were able to achieve nearly perfect scaling; see Figure 4c.

## 6 Related work

Others have proposed alternatives to mutual exclusion and coordination-free parallelism for machine learning algorithm design. [14] proposed transforming the underlying model to expose additional parallelism while preserving the marginal posterior. However, such constructions can be challenging or infeasible and many hinder mixing or convergence. Likewise, [15] proposed a reparameterization of the underlying model to expose additional parallelism through conditional independence. Additional

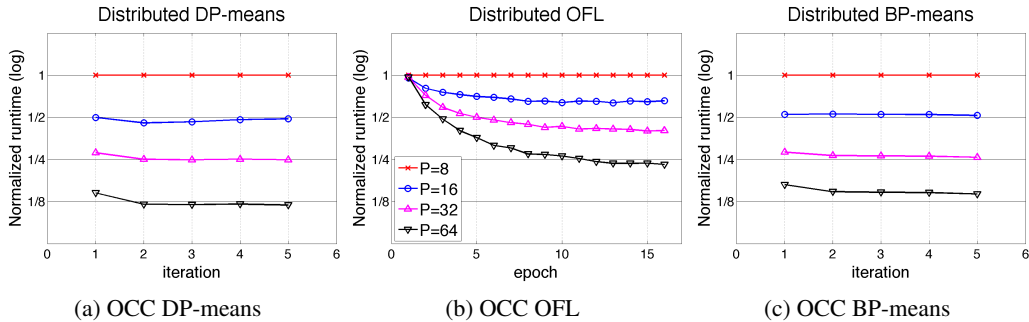

Figure 4: Normalized runtime for distributed algorithms. Runtime of each iteration / epoch is divided by that using 1 machine ($P = 8$). Ideally, the runtime with 2, 4, 8 machines ($P = 16, 32, 64$) should be respectively 1/2, 1/4, 1/8 of the runtime using 1 machine. OCC DP-means and BP-means obtain nearly perfect scaling for all iterations. OCC OFL rejects a lot initially, but quickly gets better in later epochs.

work similar in spirit to ours using OCC-like techniques includes [16] who proposed an approximate parallel sampling algorithm for the IBP which is made exact by introducing an additional Metropolis-Hastings step, and [17] who proposed a look-ahead strategy in which future samples are computed optimistically based on the likely outcomes of current samples.

There has been substantial work on scalable clustering algorithms [18, 19, 20]. Several authors [11, 21, 22, 12] have proposed streaming approximation algorithms that rely on hierarchical divide-and-conquer schemes. The approximation factors in these algorithms are multiplicative in the hierarchy and demand a careful tradeoff between communication and approximation quality which is obviated in the OCC framework. Several methods [12, 25, 21] first collect and then re-cluster a set of centers, and therefore need to communicate all intermediate centers. Our approach avoids these stages, since a center causes no rejections in the epochs after it is established: the rejection rate does not grow with $K$. Finally, the OCC framework can easily integrate and exploit many of the ideas in the cited works.

# 7 Discussion

In this paper we have shown how optimistic concurrency control can be usefully employed in the design of distributed machine learning algorithms. As opposed to previous approaches, this preserves correctness, in most cases at a small cost. We established the equivalence of our distributed OCC DP-means, OFL and BP-means algorithms to their serial counterparts, thus preserving their theoretical properties. In particular, the strong approximation guarantees of serial OFL translate immediately to the distributed algorithm. Our theoretical analysis ensures OCC DP-means achieves high parallelism without sacrificing correctness. We implemented and evaluated all three OCC algorithms on a distributed computing platform and demonstrate strong scalability in practice.

We believe that there is much more to do in this vein. Indeed, machine learning algorithms have many properties that distinguish them from classical database operations and may allow going beyond the classic formulation of OCC. In particular we may be able to partially or *probabilistically* accept non-serializable operations in a way that preserves underlying algorithm invariants. Laws of large numbers and concentration theorems may provide tools for designing such operations. Moreover, the conflict detection mechanism can be treated as a control knob, allowing us to softly switch between stable, theoretically sound algorithms and potentially faster coordination-free algorithms.

**Acknowledgments**

This research is supported in part by NSF CISE Expeditions award CCF-1139158 and DARPA XData Award FA8750-12-2-0331, and gifts from Amazon Web Services, Google, SAP, Blue Goji, Cisco, Clearstory Data, Cloudera, Ericsson, Facebook, General Electric, Hortonworks, Intel, Microsoft, NetApp, Oracle, Samsung, Splunk, VMware and Yahoo!. This material is also based upon work supported in part by the Office of Naval Research under contract/grant number N00014-11-1-0688. X. Pan's work is also supported in part by a DSO National Laboratories Postgraduate Scholarship. T. Broderick's work is supported by a Berkeley Fellowship.

## Footnotes

[1]Code will be made available at our project page https://amplab.cs.berkeley.edu/projects/ccml/.

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
