[Supplementary Material]

# A Pseudocode for OCC BP-means

**Algorithm 6:** Serial BP-means

**Input**: data $\{x_i\}_{i=1}^N$, threshold $\lambda$
Initialize $z_{i1} = 1$, $f_1 = N^{-1}\sum_i x_i$, $K = 1$
**while** *not converged* **do**
  **for** *i = 1 to N* **do**
    **for** *k = 1 to K* **do**
      Set $z_{ik}$ to minimize $\|x_i - \sum_{j=1}^K z_{ij}f_j\|_2^2$
    **if** $\|x_i - \sum_{j=1}^K z_{ij}f_{i,j}\|_2^2 > \lambda^2$ **then**
      Set $K \leftarrow K + 1$
      Create feature $f_K \leftarrow x_i - \sum_{k=1}^K z_{ik}f_j$
      Assign $z_{iK} \leftarrow 1$ (and $z_{iK} \leftarrow 0$ for $j \neq i$)
  $F \leftarrow (Z^T Z)^{-1} Z^T X$

---

**Algorithm 7:** `BPValidate`

**Input**: Set of proposed feature centers $\hat{\mathcal{C}}$
$\mathcal{C} \leftarrow \emptyset$
**for** $f^{new} \in \hat{\mathcal{C}}$ **do**
  **for** $f_{k'} \in \mathcal{C}$ **do**
    Set $z_{ik'}$ to minimize
    $\|f^{new} - \sum_{f_j \in \mathcal{C}} z_{ij}f_j\|_2^2$
  **if** $\|f^{new} - \sum_{f_j \in \mathcal{C}} z_{ij}f_j\|_2^2 > \lambda^2$ **then**
    $\mathcal{C} \leftarrow \mathcal{C} \cup \left\{ f^{new} - \sum_{f_j \in \mathcal{C}} z_{ij}f_j \right\}$
  `Ref`($f^{new}$) $\leftarrow \{z_{ij}\}_{f_j \in \mathcal{C}}$

**Output**: Accepted feature centers $\mathcal{C}$

---

**Algorithm 8:** Parallel BP-means

**Input**: data $\{x_i\}_{i=1}^N$, threshold $\lambda$
**Input**: Epoch size $b$ and $P$ processors
**Input**: Partitioning $\mathcal{B}(p,t)$ of data $\{x_i\}_{i \in \mathcal{B}(p,t)}$ to
        processor-epochs where $b = |\mathcal{B}(p,t)|$
$\mathcal{C} \leftarrow \emptyset$
**while** *not converged* **do**
  **for** *epoch t = 1 to N/(Pb)* **do**
    $\hat{\mathcal{C}} \leftarrow \emptyset$ // New candidate features
    **for** $p \in \{1, \ldots, P\}$ **do in parallel**
      // Process local data
      **for** $i \in \mathcal{B}(p,t)$ **do**
        // Optimistic Transaction
        **for** $f_k \in \mathcal{C}$ **do**
          Set $z_{ik}$ to minimize
          $\|x_i - \sum_j z_{ij}f_j\|_2^2$
        **if** $\|x_i - \sum_j z_{ij}f_j\|_2^2 > \lambda^2$ **then**
          $f_i^{new} \leftarrow x_i - \sum_j z_{ij}f_j$
          $z_i \leftarrow z_i \oplus$ `Ref`($f_i^{new}$)
          $\hat{\mathcal{C}} \leftarrow \hat{\mathcal{C}} \cup f_i^{new}$

    // Serially validate features
    $\mathcal{C} \leftarrow \mathcal{C} \cup$ DPValidate($\hat{\mathcal{C}}$)
  Compute $Z^T Z = \sum_i z_i z_i^T$ and $Z^T X = \sum_i z_i x_i^T$
  in parallel
  Re-estimate features $F \leftarrow (Z^T Z)^{-1} Z^T X$
**Output**: Accepted feature centers $\mathcal{C}$

Figure 5: The Serial BP-Means algorithm and parallel implementation using the OCC pattern, similar to OCC DP-means. Instead of proposing new clusters centered at the data point $x_i$, in OCC BP-means we propose features $f_i^{new}$ that allow us to obtain perfect representations of the data point. The validation process continues to improve on the representation $x_i \approx \sum_k z_{ik}f_k$ by using the most recently accepted features $f_{k'} \in \hat{\mathcal{C}}$, and only accepts a proposed feature if the data point is still not well-represented.

# B Proof of serializability of distributed algorithms

## B.1 Proof of Theorem 4.1 for DP-means

We note that both distributed DP-means and BP-means iterate over $z$-updates and cluster / feature means re-estimation until convergence. In each iteration, distributed DP-means and BP-means perform the same set of updates as their serial counterparts. Thus, it suffices to show that each iteration of the distributed algorithm is serially equivalent to an iteration of the serial algorithm.

Consider the following ordering on transactions:

- Transactions on individual data points are ordered before transactions that re-estimate cluster / feature means are ordered.
- A transaction on data point $x_i$ is ordered before a transaction on data point $x_j$ if
  1. $x_i$ is processed in epoch $t$, $x_j$ is processed in epoch $t'$, and $t < t'$
  2. $x_i$ and $x_j$ are processed in the same epoch, $x_i$ and $x_j$ are not sent to the master for validation, and $i < j$
  3. $x_i$ and $x_j$ are processed in the same epoch, $x_i$ is not sent to the master for validation but $x_j$ is

4. $x_i$ and $x_j$ are processed in the same epoch, $x_i$ and $x_j$ are sent to the master for validation, and the master serially validates $x_i$ before $x_j$

We show below that the distributed algorithms are equivalent to the serial algorithms under the above ordering, by inductively demonstrating that the outputs of each transaction is the same in both the distributed and serial algorithms.

Denote the set of clusters after the $t$ epoch as $\mathcal{C}^t$.

The first transaction on $x_j$ in the serial ordering has $\mathcal{C}^0$ as its input. By definition of our ordering, this transaction belongs the first epoch, and is either (1) not sent to the master for validation, or (2) the first data point validated at the master. Thus in both the serial and distributed algorithms, the first transaction either (1) assigns $x_j$ to the closest cluster in $\mathcal{C}^0$ if $\min_{\mu_k \in \mathcal{C}^0} \|x_j - \mu_k\| < \lambda$, or (2) creates a new cluster with center at $x_j$ otherwise.

Now consider any other transaction on $x_j$ in epoch $t$.

**Case 1**: $x_j$ is not sent to the master for validation.

In the distributed algorithm, the input to the transaction is $\mathcal{C}^{t-1}$. Since the transaction is not sent to the master for validation, we can infer that there exists $\mu_k \in \mathcal{C}^{t-1}$ such that $\|x_j - \mu_k\| < \lambda$.

In the serial algorithm, $x_j$ is ordered after any $x_i$ if (1) $x_i$ was processed in an earlier epoch, or (2) $x_i$ was processed in the same epoch but not sent to the master (i.e. does not create any new cluster) and $i < j$. Thus, the input to this transaction is the set of clusters obtained at the end of the previous epoch, $\mathcal{C}^{t-1}$, and the serial algorithm assigns $x_j$ to the closest cluster in $\mathcal{C}^{t-1}$ (which is less than $\lambda$ away).

**Case 2**: $x_j$ is sent to the master for validation.

In the distributed algorithm, $x_j$ is not within $\lambda$ of any cluster center in $\mathcal{C}^{t-1}$. Let $\hat{\mathcal{C}}^t$ be the new clusters created at the master in epoch $t$ before validating $x_j$. The distributed algorithm either (1) assigns $x_j$ to $\mu_{k^*} = \operatorname{argmin}_{\mu_k \in \hat{\mathcal{C}}^t} \|x_j - \mu_k\|$ if $\|x_j - \mu_k\| \leq \lambda$, or (2) creates a new cluster with center at $x_j$ otherwise.

In the serial algorithm, $x_j$ is ordered after any $x_i$ if (1) $x_i$ was processed in an earlier epoch, or (2) $x_i$ was processed in the same epoch $t$, but $x_i$ was not sent to the master (i.e. does not create any new cluster), or (3) $x_i$ was processed in the same epoch $t$, $x_i$ was sent to the master, and serially validated at the master before $x_j$. Thus, the input to the transaction is $\mathcal{C}^{t-1} \cup \hat{\mathcal{C}}^t$. We know that $x_j$ is not within $\lambda$ of any cluster center in $\mathcal{C}^{t-1}$, so the outcome of the transaction is either (1) assign $x_j$ to $\mu_{k^*} = \operatorname{argmin}_{\mu_k \in \hat{\mathcal{C}}^t} \|x_j - \mu_k\|$ if $\|x_j - \mu_k\| \leq \lambda$, or (2) create a new cluster with center at $x_j$ otherwise. This is exactly the same as the distributed algorithm.

## B.2 Proof of Theorem 4.1 for BP-means

The serial ordering for BP-means is exactly the same as that in DP-means. The proof for the serializability of BP-means follows the same argument as in the DP-means case, except that we perform feature assignments instead of cluster assignments.

## B.3 Proof of Theorem 4.1 for OFL

Here we prove Theorem 4.1 that the distributed OFL algorithm is equivalent to a serial algorithm.

*(Theorem 4.1, OFL).* We show that with respect to the returned centers (facilities), the distributed OFL algorithm is equivalent to running the serial OFL algorithm on a particular permutation of the input data. We assume that the input data is randomly permuted and the indices $i$ of the points $x_i$ refer to this permutation. We assign the data points to processors by assigning the first $b$ points to processor $p_1$, the next $b$ points to processor $p_2$, and so on, cycling through the processors and assigning them batches of $b$ points, as illustrated in Figure 6. In this respect, our ordering is generic, and can be adapted to any assignments of points to processors. We assume that each processor visits its points in

| Processor 1 | | | Processor 2 | | | | Processor $P$ | | | |
|---|---|---|---|---|---|---|---|---|---|---|
| $\mathcal{B}(1,1)$ | $\mathcal{B}(1,2)$ | ... | $\mathcal{B}(2,1)$ | $\mathcal{B}(2,2)$ | ... | ... | $\mathcal{B}(P,1)$ | $\mathcal{B}(P,2)$ | | ... |

| Serial | | | | | | | | |
|---|---|---|---|---|---|---|---|---|
| $\mathcal{B}(1,1)$ | $\mathcal{B}(2,1)$ | ... | $\mathcal{B}(P,1)$ | $\mathcal{B}(1,2)$ | $\mathcal{B}(2,2)$ | ... | ... | $\mathcal{B}(P,N/(Pb))$ |

Figure 6: Illustration of distributed and serial order of blocks $\mathcal{B}(i,t)$ of length $b$ for OFL. The order within each block is maintained. Block $\mathcal{B}(i,t)$ is processed in epoch $t$ by processor $p_i$.

the order induced by the indices, and likewise the master processes the points of an epoch in that order.

For the serial algorithm, we will use the following ordering of the data: Point $x_i$ precedes point $x_j$ if

1. $x_i$ is processed in epoch $t$ and $x_j$ is processed in epoch $t'$, and $t < t'$, or

2. $x_i$ and $x_j$ are processed in the same epoch and $i < j$.

If the data is assigned to processors as outlined above, then the serial algorithm will process the points exactly in the order induced by the indices. That means the set of points processed in any given epoch $t$ is the same for the serial and distributed algorithm. We denote by $\mathcal{C}^t$ the global set of validated centers collected by OCC OFL up to (including) epoch $t$, and by $\tilde{\mathcal{C}}^i$ the set of centers collected by the serial algorithm up to (including) point $x_i$.

We will prove the equivalence inductively.

**Epoch $t = 1$.** In the first epoch, all points are sent to the master. These are the first $Pb$ points. Since the master processes them in the same order as the serial algorithm, the distributed and serial algorithms are equivalent.

**Epoch $t > 1$.** Assume that the algorithms are equivalent up to point $x_{i-1}$ in the serial order, and point $x_i$ is processed in epoch $t$. By assumption, the set $\mathcal{C}^{t-1}$ of global facilities for the distributed algorithm is the same as the set $\tilde{\mathcal{C}}^{(t-1)Pb}$ collected by the serial algorithm up to point $x_{(t-1)Pb}$. For notational convenience, let $D(x_i, \mathcal{C}^t) = \min_{\mu \in \mathcal{C}^t} \|x_i - \mu\|^2$ be the distance of $x_i$ to the closest global facility.

The essential issue to prove is the following claim:

**Claim 1.** *If the algorithms are equivalent up to point $x_{i-1}$, then the probability of $x_i$ becoming a new facility is the same for the distributed and serial algorithm.*

The serial algorithm accepts $x_i$ as a new facility with probability $\min\{1, D(x_i, \tilde{\mathcal{C}}^{i-1})/\lambda^2\}$. The distributed algorithm sends $x_i$ to the master with probability $\min\{1, D(x_i, \mathcal{C}^{t-1})/\lambda^2\}$. The probability of ultimate acceptance (validation) of $x_i$ as a global facility is the probability of being sent to the master *and* being accepted by the master. In epoch $t$, the master receives a set of candidate facilities with indices between $(t-1)Pb + 1$ and $tPb$. It processes them in the order of their indices, i.e., all candidates $x_j$ with $j < i$ are processed before $i$. Hence, the assumed equivalence of the algorithms up to point $x_{i-1}$ implies that, when the master processes $x_i$, the set $\mathcal{C}^{t-1} \cup \hat{\mathcal{C}}$ equals the set of facilities $\tilde{\mathcal{C}}^{i-1}$ of the serial algorithm. The master consolidates $x_i$ as a global facility with probability 1 if $D(x_i, \tilde{\mathcal{C}}^{i-1} \cup \hat{\mathcal{C}}) > \lambda^2$ and with probability $D(x_i, \tilde{\mathcal{C}}^{i-1} \cup \hat{\mathcal{C}})/D(x_i, \mathcal{C}^{t-1})$ otherwise.

We now distinguish two cases. If the serial algorithm accepts $x_i$ because $D(x_i, \tilde{\mathcal{C}}^{i-1}) \geq \lambda^2$, then for the distributed algorithm, it holds that

$$D(x_i, \mathcal{C}^{t-1}) \geq D(x_i, \mathcal{C}^{t-1} \cup \hat{\mathcal{C}}) = D(x_i, \tilde{\mathcal{C}}^{i-1}) \geq \lambda^2 \qquad (1)$$

and therefore the distributed algorithm also always accepts $x_i$.

Otherwise, if $D(x_i, \tilde{\mathcal{C}}^{i-1}) < \lambda^2$, then the serial algorithm accepts with probability $D(x_i, \tilde{\mathcal{C}}^{i-1})/\lambda^2$. The distributed algorithm accepts with probability

$$\mathbb{P}(x_i \text{ accepted}) = \mathbb{P}(x_i \text{ sent to master }) \cdot \mathbb{P}(x_i \text{ accepted at master} \mid x_i \text{ sent}) \tag{2}$$

$$= \frac{D(x_i, \mathcal{C}^{t-1})}{\lambda^2} \cdot \frac{D(x_i, \tilde{\mathcal{C}}^{i-1} \cup \hat{\mathcal{C}})}{D(x_i, \mathcal{C}^{t-1})} \tag{3}$$

$$= \frac{D(x_i, \tilde{\mathcal{C}}^{i-1})}{\lambda^2}. \tag{4}$$

This proves the claim.

The claim implies that if the algorithms are equivalent up to point $x_{i-1}$, then they are also equivalent up to point $x_i$. This proves the theorem. $\qquad\square$

### B.4    Proof of Theorem 4.2 (Approximation bound)

We begin by relating the results of facility location algorithms and DP-means. Recall that the objective of DP-means and FL is

$$J(\mathcal{C}) = \sum_{x \in X} \min_{\mu \in \mathcal{C}} \|x - \mu\|^2 + \lambda^2 |\mathcal{C}|. \tag{5}$$

In FL, the facilities may only be chosen from a pre-fixed set of centers (e.g., the set of all data points), whereas DP-means allows the centers to be arbitrary, and therefore be the empirical mean of the points in a given cluster. However, choosing centers from among the data points still gives a factor-2 approximation. Once we have established the corresponding clusters, shifting the means to the empirical cluster centers never hurts the objective. The following proposition has been a useful tool in analyzing clustering algorithms:

**Proposition B.1.** *Let $\mathcal{C}^*$ be an optimal solution to the DP-means problem* (5)*, and let $\mathcal{C}^{FL}$ be an optimal solution to the corresponding FL problem, where the centers are chosen from the data points. Then*

$$J(\mathcal{C}^{FL}) \le 2J(\mathcal{C}^*).$$

*Proof. (Proposition B.1)* It is folklore that Proposition B.1) holds for the K-means objective, i.e.,

$$\min_{\mathcal{C} \subseteq X, |\mathcal{C}|=k} \sum_{i=1}^{n} \min_{\mu \in \mathcal{C}} \|x_i - \mu\|^2 \le 2 \min_{\mathcal{C} \subseteq X} \sum_{i=1}^{n} \min_{\mu \in \mathcal{C}} \|x_i - \mu\|^2. \tag{6}$$

In particular, this holds for the optimal number $K^* = |\mathcal{C}^*|$. Hence, it holds that

$$J(\mathcal{C}^{FL}) \le \min_{\mathcal{C} \subseteq X, |\mathcal{C}|=K^*} \sum_{i=1}^{n} \min_{\mu \in \mathcal{C}} \|x_i - \mu\|^2 + \lambda^2 K^* \le 2J(\mathcal{C}^*). \tag{7}$$

$\qquad\square$

With this proposition at hand, all that remains is to prove an approximation factor for the FL problem.

*Proof. (Theorem 4.2)* First, we observe that the proof of Theorem 4.1 implies that, for any random order of the data, the OCC and serial algorithm process the data in exactly the same way, performing ultimately exactly the same operations. Therefore, any approximation factor that holds for the serial algorithm straightforwardly holds for the OCC algorithm too.

Hence, it remains to prove the approximation factor of the serial algorithm. Let $C_1^{\text{FL}}, \ldots, C_k^{\text{FL}}$ be the clusters in an optimal solution to the FL problem, with centers $\mu_1^{\text{FL}}, \ldots \mu_k^{\text{FL}}$. We analyze each optimal cluster individually. The proof follows along the lines of the proofs of Theorems 2.1 and 4.2 in [10], adapting it to non-metric squared distances. We show the proof for the constant factor, the logarithmic factor follows analogously by using the ring-splitting as in [10].

First, we see that the expected total cost of any point $x$ is bounded by the distance to the closest open facility $y$ that is present when $x$ arrives. If we always count in the distance of $\|x - y\|^2$ into the cost of $x$, then the expected cost is $\gamma(x) = \lambda^2 \|x - y\|^2 / \lambda^2 + \|x - y\|^2 = 2\|x - y\|^2$.

We consider an arbitrary cluster $C_i^*$ and divide it into $|C^*|/2$ *good* points and $|C^*|/2$ *bad points*. Let $D_i = \frac{1}{|C^{\text{FL}}|}\sum_{x \in C_i^*}\|x - \mu_i\|$ be the average service cost of the cluster, and let $d_g$ and $d_b$ be the service cost of the good and bad points, respectively (i.e., $D_i = (d_g + d_b)/|C_i^{\text{FL}}|$). The good points satisfy $\|x - \mu_i^{\text{FL}}\| \le 2D_i$. Suppose the algorithm has chosen a center, say $y$, from the points $C_i^{\text{FL}}$. Then any other point $x \in C_i^{\text{FL}}$ can be served at cost at most

$$\|x - y\|^2 \le \left(\|x - \mu_i^{\text{FL}}\| + \|y - \mu_i^{\text{FL}}\|\right)^2 \le 2\|x - \mu_i^{\text{FL}}\|^2 + 4D_i. \tag{8}$$

That means once the algorithm has established a good center within $C_i^{\text{FL}}$, all other good points together may be serviced within a constant factor of the total optimal service cost of $C^{\text{FL}}$, i.e., at $2d_g + 4(d_g + d_b)$. The assignment cost of all the good points in $C_i^{\text{FL}}$ that are passed before opening a good facility is, by construction of the algorithm and expected waiting times, in expectation $\lambda^2$. Hence, in expectation, the cost of the good points in $C_i^{\text{FL}}$ will be bounded by $\sum_{x\text{good}}\gamma(x) \le 2(2d_g + 4d_g + 4d_b + \lambda^2)$.

Next, we bound the expected cost of the bad points. We may assume that the bad points are injected randomly in between the good points, and bound the servicing cost of a bad point $x_b \in C_i^{\text{FL}}$ in terms of the closest good point $x_g \in C_i^{\text{FL}}$ preceding it in our data sequence. Let $y$ be the closest open facility to $\mu_i^{\text{FL}}$ when $y$ arrives. Then

$$\|x_b - y\|^2 \le 2\|y - \mu_i^{\text{FL}}\|^2 + 2\|x_b - \mu^{\text{FL}}\|^2. \tag{9}$$

Now assume that $x_g$ was assigned to $y'$. Then

$$\|y - \mu_i^{\text{FL}}\|^2 \le \|y' - \mu_i^{\text{FL}}\|^2 \le 2\|y' - x_g\|^2 + 2\|x_g - \mu^{\text{FL}}\|^2. \tag{10}$$

From (9) and (8), it then follows that

$$\|x_b - y\|^2 \le 4\|y' - x_g\|^2 + 4\|x_g - \mu^{\text{FL}}\|^2 + 2\|x_b - \mu^{\text{FL}}\|^2 \tag{11}$$

$$= 2\gamma(x_g) + 4\|x_g - \mu^{\text{FL}}\|^2 + 2\|x_b - \mu^{\text{FL}}\|^2. \tag{12}$$

Since the data is randomly permuted, $x_g$ could be, with equal probability, any good point, and in expectation we will average over all good points.

Finally, with probability $2/|C_i^{\text{FL}}|$ there is no good point before $x_g$. In that case, we will count in $x_b$ as the most costly case of opening a new facility, incurring cost $\lambda^2$. In summary, we can bound the expected total cost of $C^{\text{FL}}$ by

$$\sum_{x \text{ good}}\gamma(x) + \sum_{x \text{ bad}}\gamma(x) \le 12d_g + 8d_b + \lambda^2 + \frac{2C^{\text{FL}}}{2C^{\text{FL}}}\lambda^2 + 2(2\frac{2|C_i^{\text{FL}}|}{2|C^{\text{FL}}|}(12d_g + 8d_b + \lambda^2) + 4d_g + 2d_b)$$

$$\tag{13}$$

$$\le 68d_g + 42d_b + 4\lambda^2 \le 68J(\mathcal{C}^{\text{FL}}). \tag{14}$$

This result together with Proposition B.1 proves the theorem. □

## C   Master processing bound for DP-means (Theorem 4.3)

We restate Theorem 4.3 here for convenience.

**DP-Means Scalability.** Assume $N$ data points are generated iid to form a random number $(K_N)$ of well-spaced clusters of diameter $\lambda$: $\lambda$ is an upper bound on the distances within clusters and a lower bound on the distance between clusters. Then the expected number of *serially* validated points is bounded above by $Pb + \mathbf{E}[K_N]$ for $P$ processors and $b$ points per epoch.

*Proof.* As in the theorem statement, we assume $P$ processors, $b$ points assigned to each processor per epoch, and $N$ total data points. We further assume a generative model for the cluster memberships: namely, that they are generated iid from an arbitrary distribution $(\pi_j)_{j=1}^\infty$. That is, we have $\sum_{j=1}^\infty \pi_j = 1$ and, for each $j$, $\pi_j \in [0, 1]$. We see that there are perhaps infinitely many latent clusters. Nonetheless, in any data set of finite size $N$, there will of course be only finitely many clusters to which any data point in the set belongs. Call the number of such clusters $K_N$.

Consider any particular cluster indexed by $j$. At the end of the first epoch in which a worker sees $j$, that worker (and perhaps other workers) will send some data point from $j$ to the master. By construction, some data point from $j$ will belong to the collection of cluster centers at the master by the end of the processing done at the master and therefore by the beginning of the next epoch. It follows from our assumption (all data points within a single cluster are within a $\lambda$ diameter) that no other data point from cluster $j$ will be sent to the master in future epochs. It follows from our assumption about the separation of clusters that no points in other clusters will be covered by any data point from cluster $j$.

Let $S_j$ represent the (random) number of points from cluster $j$ sent to the master. Since there are $Pb$ points processed by workers in a single epoch, $N_j$ is constrained to take values between $0$ and $Pb$. Further, note that there are a total of $N/(Pb)$ epochs.

Let $A_{j,s,t}$ be the event that the master is sent $s$ data points from cluster $j$ in epoch $t$. All of the events $\{A_{j,s,t}\}$ with $s = 1, \ldots, Pb$ and $t = 1, \ldots, N/(Pb)$ are disjoint. Define $A'_{j,0}$ to be the event that, for all epochs $t = 1, \ldots, N/(Pb)$, zero data points are sent to the master; i.e., $A'_{j,0} := \bigcup_t A_{j,0,t}$. Then $A'_{j,0}$ is also disjoint from the events $\{A_{j,s,t}\}$ with $s = 1, \ldots, Pb$ and $t = 1, \ldots, N/(Pb)$. Finally,

$$A'_{j,0} \cup \bigcup_{s=1}^{Pb} \bigcup_{t=1}^{N/(Pb)} A_{j,s,t}$$

covers all possible data configurations. It follows that

$$\mathbb{E}[S_j] = 0 * \mathbb{P}[A'_{j,0}] + \sum_{s=1}^{Pb} \sum_{t=1}^{N/(Pb)} s\mathbb{P}[A_{j,s,t}] = \sum_{s=1}^{Pb} \sum_{t=1}^{N/(Pb)} s\mathbb{P}[A_{j,s,t}]$$

Note that, for $s$ points from cluster $j$ to be sent to the master at epoch $t$, it must be the case that no points from cluster $j$ were seen by workers during epochs $1, \ldots, t-1$, and then $s$ points were seen in epoch $t$. That is, $\mathbb{P}[A_{j,s,t}] = (1 - \pi_j)^{Pb(t-1)} \cdot \binom{Pb}{s} \pi_j^s (1 - \pi_j)^{Pb-s}$.

Then

$$\mathbb{E}[S_j] = \left( \sum_{s=1}^{Pb} s \binom{Pb}{s} \pi_j^s (1 - \pi_j)^{Pb-s} \right) \cdot \left( \sum_{t=1}^{N/(Pb)} (1 - \pi_j)^{Pb(t-1)} \right)$$

$$= \pi_j Pb \cdot \frac{1 - (1 - \pi_j)^{Pb \cdot N/(Pb)}}{1 - (1 - \pi_j)^{Pb}},$$

where the last line uses the known, respective forms of the expectation of a binomial random variable and of the sum of a geometric series.

To proceed, we make use of a lemma.

**Lemma C.1.** *Let $m$ be a positive integer and $\pi \in (0, 1]$. Then*

$$\frac{1}{1 - (1 - \pi)^m} \leq \frac{1}{m\pi} + 1.$$

*Proof.* A particular subcase of Bernoulli's inequality tells us that, for integer $l \leq 0$ and real $x \geq -1$, we have $(1 + x)^l \geq 1 + lx$. Choose $l = -m$ and $x = -\pi$. Then

$$(1 - \pi)^m \leq \frac{1}{1 + m\pi}$$

$$\Leftrightarrow 1 - (1 - \pi)^m \geq 1 - \frac{1}{1 + m\pi} = \frac{m\pi}{1 + m\pi}$$

$$\Leftrightarrow \frac{1}{1 - (1 - \pi)^m} \leq \frac{m\pi + 1}{m\pi} = \frac{1}{m\pi} + 1.$$

$\square$

We can use the lemma to find the expected total number of data points sent to the master:

$$\mathbb{E}\sum_{j=1}^{\infty} S_j = \sum_{j=1}^{\infty} \mathbb{E}S_j = \sum_{j=1}^{\infty} \pi_j Pb \cdot \frac{1 - (1 - \pi_j)^N}{1 - (1 - \pi_j)^{Pb}}$$

$$\leq \sum_{j=1}^{\infty} \pi_j Pb \cdot \left(1 + \frac{1}{\pi_j Pb}\right) \cdot \left(1 - (1 - \pi_j)^N\right)$$

$$= Pb \sum_{j=1}^{\infty} \pi_j \left(1 - (1 - \pi_j)^N\right) + \sum_{j=1}^{\infty} \left(1 - (1 - \pi_j)^N\right)$$

$$\leq Pb + \sum_{j=1}^{\infty} \mathbb{P}(\text{cluster } j \text{ occurs in the first } N \text{ points})$$

$$= Pb + \mathbb{E}[K_N].$$

Conversely,

$$\mathbb{E}\sum_{j=1}^{\infty} S_j \geq \sum_{j=1}^{\infty} \pi_j Pb = Pb.$$

$\square$

To analyze the total running time, we note that after each of the $N/(Pb)$ epochs the master and workers must communicate. Each worker must process $N/P$ data points, and the master sees at most $k_N + Pb$ points. Thus, the total expected running time is $O(N/(Pb) + N/P + Pb)$.

### C.1 Experiment – under assumptions of theorem

To demonstrate the bound on the expected number of data points proposed but not accepted as new centers, we generated synthetic data with separable clusters. Cluster proportions are generated using the stick-breaking procedure for the Dirichlet process, with concentration parameter $\theta = 1$. Cluster means are set at $\mu_k = (2k, 0, 0, \ldots, 0)$, and generated data uniformly in a ball of radius $1/2$ around each center. Thus, all data points from the same cluster are at most distance 1 from one another, and more than distance of 1 from any data point from a different cluster.

We follow the same experimental framework in Section 5.1.

(a) DP-means, separable    (b) OFL, separable

Figure 7: Simulated distributed DP-means and OFL: expected number of data points proposed but not accepted as new clusters is independent of size of data set.

In the case where we have separable clusters (Figure 7), $\hat{\mathbb{E}}[M_N - k_N]$ is bounded from above by $Pb$, which is in line with the above Theorem 4.3.

## C.2 Experiment – assumptions of theorem violated

We further examine the rejection rate of distributed DP-means when the assumptions of the theorem are violated. Specifically, we vary the separation between cluster centers, and the algorithm choice of $\lambda$ relative to the cluster diameter.

Data is synthetically generated using the stick-breaking process with $\theta = 1$, truncated to a maximum of 16 clusters in 16 dimensions. Cluster centers are chosen to keep all pairwise distances at a fixed separation. Data points are generated in balls of radius 1 around the cluster centers.

Separation between cluster centers is varied between 0 (completely overlapping), 1 (partially overlapping), and 2 (almost separated). The choice of $\lambda$ is varied between 0.5 (smaller than actual radius), 1, 2, and 4. Note that data points from the same cluster are within a distance of 2 from one another, and all data points are within a distance of 4 from one another.

Figure 8 shows the number of cluster centers that are rejected by distributed DP-means. We observe that the bounds of the theorem hold even when the assumptions are violated.

(a) $\lambda = 0.5$, separation 0      (b) $\lambda = 0.5$, separation 1      (c) $\lambda = 0.5$, separation 2

(d) $\lambda = 1$, separation 0      (e) $\lambda = 1$, separation 1      (f) $\lambda = 1$, separation 2

(g) $\lambda = 2$, separation 0      (h) $\lambda = 2$, separation 1      (i) $\lambda = 2$, separation 2

(j) $\lambda = 4$, separation 0      (k) $\lambda = 4$, separation 1      (l) $\lambda = 4$, separation 2

Figure 8: Number of proposed but rejected cluster centers, for various choices of $\lambda$ and underlying separation between cluster centers. Note that the y-axis for $\lambda = 0.5$ is different than for other choices of $\lambda$.