[Reviews · NeurIPS 2013]

Submitted by Assigned_Reviewer_4

The author essentially tried to propose parallel algorithms for a clustering method (OFL as an extension of DP-means) and a feature selection method. These algorithm is then unified into optimistic concurrency control that was introduced in parallel database community.

It seems that the algorithm that the author proposed can be perfectly fitted into the mapreduce framework. The map stage is the parallel operation, and the reduce stage is the serial operation.

The theoretical analysis provides little added value, as they are mostly straight-forward. The experiment is not impressive because with speedup is only 4x on 8 processors.

--- update---

The problem of the paper is obvious: too ad-hoc algorithms for the demonstration and poor performance evaluation. If the paper could position itself as a parallelized DP-means algorithm with extended discussion on OCC, it could be a better paper.
Summary: Algorithm too ad-hoc and the performance of the algorithms is questionable.

Submitted by Assigned_Reviewer_5

Running machine learning algorithms on larger datasets is becomming more and more a necessity. Recently, a practically very relevant line of research has been to look at various programming paradigms for turning well known machine learning algorithms into distributed algorithms - meaning they can run on an infrastructure with no shared memory and slow communication between processing units.

This paper introduces a well known pattern called "optimistic concurrency control" into the machine learning literature. As the authors point out, there has been some work on embarrasingly parallel algorithms, distributed algorithm using the locking paradigm and coordination-free approaches to distributed algorithms. Optimistic concurrency control is a technique which starts out by assuming that each individual processing unit can freely access shared state. At certain points in the computation the algorithms checkpoint and verify the assumption didn't harm the correctness; if the assumption was deemed incorrect, part of the computation is rolled back or corrected.

The authors make a convincing case that optimistic concurrency control (OCC henceforth) is worthwhile investigating in the machine learning context. They go on to apply OCC to three algorithms DP-means, the Facility Location problem and BP-means. I found the explanation of the algorithm very clear and understandable. A very minor point that might deserve a mention in the appendix is how the first clusters are assigned (i.e. how do we argmin_{\mu \in C} when C is empty)?

Reading through Algorithm 3, one thing I wondered about is how expensive it is for a distributed algorithm to be sending x_i between different processing units? I'm not sure how to solve this problem, but in my experience moving the dataset around the cluster is more expensive than moving the parameters around the cluster. I'd love to see some discussion on this point.

The authors continue to discuss correctness proofs for the algorithms. I've only skimmed the proofs in the appendix but they seem correct. In the evaluation the authors first run the algorithms on a small synthetic dataset to drive the point home that the number of corrections that need to be made is modest. In a larger experiment using Spark on Amazon AWS, the authors show convincing results on large datasets. It wasn't clear to me what data was used for this experiment (synthetic?).


line 33: iid => i.i.d.
Summary: A well written paper on how to use optimistic concurrency control to implement distributed machine learning algorithms.

Submitted by Assigned_Reviewer_6

This paper studies the problem of large scale unsupervised learning. It proposes the paradigm of “optimistic concurrency control”, which assumes that conflicts are unlikely and if conflicts do arise a conflict-resolution protocol is invoked. The paper then applies the approach to three clustering related problems: clustering, feature learning and online facility location. It also proves the serializability of the proposed algorithms. It provides the experimental results on simulated data sets.

This paper is well-organized and clearly written. It provides enough technical details and theoretical support. The idea of OOC is neat and practical. Although the paper presents the idea from the point view of concurrency control, however in more general speaking, the key point of large scale machine learning is how to update the global parameter from the local parameters from each parallelized parts. The proposed approach provides a simple and practical way to update the global parameters (global cluster assignments) from local ones (local cluster assignments). However, this update strategy does not necessarily work for other unsupervised learning or supervised learning algorithms. In general, the update strategy is algorithm-dependent. My major concern about this work is its experimental results. The experiments cannot prove the usefulness of the proposed algorithms. First, it is only based on simulated data, which does not provide insights how the algorithm will be useful for real data sets. Second, it should provide cluster quality comparison with other distributed clustering algorithms in the literature.
Summary: The paper provides a neat idea for distribute unsupervised learning (clustering) with a good presentation. However, the experiments are not strong enough to support the usefulness of the proposed algorithms.
Author Feedback

Author rebuttal: Assigned_Reviewer_4

MapReduce Comparison
1. Our paper proposes the OCC framework for parallelizing serial algorithms that involve many small updates iteratively transforming a global state, a task that MapReduce is ill-suited for. To achieve this, the OCC pattern introduces a validation operation that can possibly invoke a rejection or rollback, which is typically not allowed in the reduce phase. Furthermore, as mentioned in Section 2.1, OCC transactions can occur asynchronously, resulting in greater potential speedup. Admittedly, the presentation of the algorithms in a batch-wise form would appear to be similar to the Mapreduce pattern; however, this is only done for ease of presentation and analysis.

Theoretical Analysis
2.1. A big advantage of the proposed approach is that theoretical analyses with respect to correctness and approximation bounds become simpler. (Other distributed clustering algorithms, in contrast, require a far more tedious analysis.) A more machine-learner-friendly framework is one of the contributions of this paper.

2.2. Furthermore, in general, the most important part of a theorem is its implications, and not the difficulty of proving it. Our theorems imply that (1) the OCC algorithms are provably correct, and (2) the overhead is _independent_ of the size of the data set. Hence, the algorithms are very well suited for large data.

Experiments --- Speedup
3. We would like to point out that DP-means, OFL, and BP-means, in their original form, are inherently serial algorithms, since each update (or transaction) depends on the outcome of the previous update. Because of this serial dependence, we should expect no speedup at all. We were, however, able to achieve speedups of almost 8x (for DP-means and BP-means) and 4x (for OFL) on 8 machines for algorithms that are, as we show, exactly equivalent to the serial algorithms.


Assigned_Reviewer_5

Initial Clusters
4. When the set of clusters, C, is initially empty, we consider the distance to be infinity, so that the next point is proposed as a new cluster. We are currently investigating alternative schemes.

Communication Cost
5. For the clustering and feature modeling problems we examined, the parameters are cluster centers and features, which are in fact represented by selected data points themselves, or are of the same size as the data points. Thus, the communication cost of moving a small fraction of the data or the parameters is the same.

Synthetic Data
6. The data used for the scaling experiments on Amazon EC2 was synthetic. We considered using real-world datasets but given the limited space we decided to focus on synthetic data which matches the DP-means modeling assumptions eliminating the need to tune the underlying models for the data (which was not a focus of this work).


Assigned_Reviewer_6

Applicability / Adaptability of OCC
7. The main message of the paper is to introduce OCC for algorithms that sequentially update global parameters (as many machine learning algorithms do), and to illustrate its main ideas and properties. The limited space forced us to restrict the presentation to the three examples. We are currently working on other algorithms. Using OCC for different algorithms indeed needs different specifications. But, that said, popular ideas like MapReduce or variational inference too need to be specified for each setting separately, and have still proven to be extremely useful for a variety of settings.

Experimental Validation --- Clustering Quality
8.1. Our paper is focused on demonstrating the simplicity and power of the OCC pattern in Machine Learning and rely on the DP-means, OFL, and BP-means algorithms as case studies where no previous distributed algorithms had been developed. The clustering quality of these algorithms have been tested in the cited works. Since our distributed variants return equivalent solutions, the previous quality studies apply to our algorithms as well.

8.2. We do not claim that our variants give qualitatively better solutions that the corresponding sequential algorithms, but we claim and show that they allow to compute equivalent solutions faster and on much larger data sets. For the questions we study, the tested data sets are suitable.